

# An English course practice evaluation system based on multi-source mobile information and IoT technology

Zhenlong Wang

Sanmenxia Polytechnic, Sanmenxia, China

## ABSTRACT

With the increased use of online English courses, the quality of the course directly determines its efficacy. Recently, various industries have continuously employed Internet of Things (IoT) technology, which has considerable scene adaptability. To better supervise the specific content of English courses, we discuss how to apply multi-source mobile Internet of Things information technology to the practical evaluation system of English courses to boost the performance of English learning evaluation. Therefore, by analyzing the problems of existing English course evaluation and the characteristics of multi-source mobile Internet of Things information technology, this article designs an English course practical evaluation system based on multi-source data collection, processing, and analysis. The system can collect real-time student voices, behavior, and other data through mobile devices. Then, analyze the data using cloud computing and data mining technology and provide real-time learning progress and feedback. We can demonstrate that the accuracy of the evaluation system can reach 80.23%, which can effectively improve the efficiency of English learning evaluation, provide a new method for English teaching evaluation, and further improve and optimize the English education teaching content to meet the needs of the actual teaching environment.

Corresponding author
Zhenlong Wang,
wangzhenlong@smxpt.edu.cn

## INTRODUCTION

Mobile Internet of Things information technology (Miot) is a new type of IoT technology (*Atzori, Iera & Morabito, 2010*; *Xumin et al., 2022*), which is real-time, reliable, and traceable and can realize seamless connections between different devices in the IoT system. Miot technology can significantly improve people's daily lives, improve production efficiency, reduce energy consumption, and realize automatic monitoring and control to meet people's growing needs. Miot technology can also be used in the practical evaluation of English courses to make students' practical evaluations more accurate, standardized, and effective.

Currently, research on mobile Internet of Things technology mainly focuses on wireless sensor networks, mobile positioning technology, mobile Internet of Things devices, and mobile Internet of Things applications. The specific research content includes the network structure of wireless sensor networks, the accurate positioning of mobile positioning

technology, the efficient energy saving of Miot devices, and the real-time monitoring of Miot applications. Many countries and regions have begun to devote themselves to the research of Miot technology and have received specific results. But it also has some problems, for instance, with the accuracy of mobile positioning technology and the security of wireless sensor networks.

In the field of English education, evaluation is the process of judging whether the learning experience has achieved the expected educational objectives; however, evaluation also involves the identification of the pros and cons of the curriculum design (*Chen, Hongtao & Jun, 2021*; *Wang, Wang & Han, 2013*; *Momirski, 2019*). Therefore, English education evaluation not only represents the degree of achievement of educational goals but also represents a method of comparison between the concept and the actual results of the plan of the educational program or the educational activity. Teaching quality can be reviewed from the perspective of teaching and the evaluation of online courses based on artificial intelligence. The problem of teaching quality involves all aspects of online course teaching. In terms of time series, it requires lesson preparation and in-class teaching. In terms of specific affairs, pre-class practice includes learning situation analysis, a teaching plan, teaching content, *etc*. In the course of teaching, it involves teaching language, teaching skills and experience, real-time monitoring, real-time feedback, and response. Therefore, evaluating English course practice synthesizes audio, video, and text generated in education. Considering the above characteristics, this article integrates multi-source mobile Internet of Things information technology to comprehensively evaluate the audio, video, and text information generated in education. This article constructs an evaluation system for English course practice using the multi-source mobile Internet of Things environment, including audio and video sensors, data service centers, *etc*. By embedding the deep learning algorithm in the multi-source mobile Internet of Things, intelligent analysis of the lesson preparation content and comprehensive scoring of the teaching process are realized.

This article aims to study the practical evaluation system of English courses integrating multi-source Miot information technology to achieve higher efficiency, higher quality, and higher coverage of English courses and provide a more effective evaluation system. Therefore, this article proposes an English course practice evaluation system based on multi-source data collection, processing, and analysis to improve English teaching reform. We will introduce the related works about our method in "Related Works", the proposed method in "The English Curriculum Practice Evaluation System Integrating Multi-Source MIoT Information Technology", and the experiments in "Experiment and Analysis".

1) The main contributions are as follows:

2) Based on Miot, we designed the English curriculum implementation evaluation system to realize the interconnection of various elements in the English curriculum practice evaluation system and maximize the convenience of Miot.

3) We proposed an evaluation method for English lesson preparation content based on intelligent text analysis to determine the quality of the prepared material.

4) We proposed an intelligent scoring method for teaching lessons to achieve the best English teaching practice performance.

## RELATED WORKS

### Research on intelligent text analysis methods

Some early textual entailment methods, such as those based on text similarity and those based on logical calculus, are too shallow to adapt to more complex semantics. The convolutional neural network (CNN)-based textual entailment model can automatically explore the N-gram features in the sentence, which has a good ability for local information mining, and the recurrent neural network (RNN)-based textual entailment model can analyze the long-term dependence and short-term dependence in the document sentence. After combining the two types of models with the attention mechanism, the discovery of the key content in the document sentence has been further improved.

Three types of models can be used to solve the problem of text understanding: attention-based models, pointer-wise models, and multi-hop mechanism models. An attention mechanism is characteristic of attention-based models. *Hermann et al. (2015)* proposed an attentive radar based on a connected attention mechanism, and *Chen, Bolton & Manning (2016)* proposed Stanford AR based on a bilinear attention mechanism. The model based on pointer-wise tries to decrease the model's perplexity by narrowing the answer's value range from the whole dictionary to a single discourse dictionary. The representatives of this model include AS Reader (*Kadlec et al., 2016*) and AOA Reader (*Cui et al., 2016*). The model simulates the multi-step reading of human beings and filters the essential parts layer by layer through the method of superimposed attention mechanism. The representatives of this type of model are AMRNN (Attention-based Multi-hop Recurrent Neural Network) (*Tseng et al., 2016*) and GAReader (*Dhingra et al., 2017*). 2018, Microsoft released the MS-MARCO dataset (*Nguyen et al., 2016*), a fragment synthesis dataset.

Meanwhile, the original version of Du Reader (*He et al., 2017*) is a complete fragment synthesis task consisting of multiple passages, an actual user's question in a Baidu search, and then a synthetic answer. This type of reading comprehension problem corresponds to certain heuristics. For example, it directly borrows the idea of extraction synthesis (S-Net (*Tan et al., 2017*)). There is also a novel read-then-verify model (*Hu et al., 2018*), which tries to construct a model containing two modules through read-verify.

### Research on intelligent scoring methods for teaching

With the popularity of online intelligent scoring systems in modern teaching, more and more scholars have begun to join the ranks of related research. First of all, regarding the reliability and validity of the intelligent scoring system, the online intelligent scoring system has the advantages of immediacy, reliability, and interactivity, and the automatic scoring system can objectively improve the efficiency of feedback. Many scholars have a positive attitude toward the validity of the intelligent scoring system (*Bull & McKenna, 2004*). *Wilson (2016)* also conducted a study about the effective feedback of intelligent

systems. He conducted a one-semester experiment on learners in lower grades and affirmed the effectiveness of intelligent feedback in assisting teachers in their feedback work.

Compared with teacher feedback, the feedback of an intelligent essay scoring system is more objective and efficient than teacher feedback, and it can better overcome the reliability and validity problems caused by some human factors, such as fatigue (*Group, 2022*). *Reilly et al. (2014)* and other researchers have also confirmed that there are differences between intelligent scoring systems and teacher feedback. In terms of the reliability and validity of the system, scholars generally think that it has the characteristics of timely feedback, is objective and effective, and has a positive attitude; they mainly believe that the system can be used as an auxiliary tool for teachers to make up for the lack of teacher feedback. However, the research on comparing teacher and machine feedback is not very comprehensive.

The online intelligent scoring system also has shortcomings in evaluating teachers' emotions in courses. Firstly, some scholars believe that the system can only play a role in supervising course fluency, and whether it is conducive to supervising students' teaching efficiency remains to be explored (*Wang & Chen, 2017*). Secondly, the feedback focuses on video and audio, which is not helpful for students' psychological acceptance. However, such problems will be reduced with the update and improvement of the system. In the current situation of the online intelligent scoring system, the shortcomings in error recognition have restricted its wide application to a certain extent and reduced the quality of students' modification feedback, so the system urgently needs improvement. Some scholars have proposed many ways to assist the course through Internet of Things technology. *Zong, Jia & Zhang (2014)* apply the IoT to managing sources in the school, such as students, the teaching environment, the library, and teaching instruments and equipment. *Qi & Shen (2011)* propose an application scheme based on the Internet of Things to solve the access defects and system security problems of RFID tags and improve the efficiency of traditional teaching management system methods. *Chen & Huang (2021)* designed an interactive online English teaching system based on Internet of Things technology and evaluated the teaching quality of the system using a gray correlation analysis algorithm.

Looking at the above content, there have been many relevant experimental studies on the intelligent teaching scoring system, and the system has also been widely used in schools at all stages. Most researchers have a positive attitude toward reliability and validity, and the research shows the system can improve the efficiency of students' courses. Of course, some scholars think the system has limitations and defects in the teacher's mental outlook.

## THE ENGLISH CURRICULUM PRACTICE EVALUATION SYSTEM INTEGRATING MULTI-SOURCE MIOT INFORMATION TECHNOLOGY

In the IoT, many sensors and terminal devices scattered in different locations cooperate to form a comprehensive coverage of the IoT perception network and transmit data to support various applications of the Internet of Things. The IoT first collects information

with a variety of sensors. Then, the collected information is communicated accurately in real time based on adapting to various heterogeneous networks and protocols. Finally, combined with intelligent processing technology, the collected data is mined and analyzed to meet the needs of different users. The English course practice evaluation system integrating multi-source mobile Internet of Things technology is shown in Fig. 1.

The perception layer of the English curriculum implementation evaluation system is the basic core part, which is composed of a variety of sensor devices, multimedia acquisition devices, radio frequency identification devices (RFID), smart devices, *etc.*, and is used to collect relevant data and physical information about the world. The perception layer mainly establishes a connection with other nearby devices, collaboratively forms a network, receives appropriate control commands and function commands, realizes intelligent perception recognition, information collection, and processing, and the communication module sends the collected data to the network layer. The perception layer mainly uses a wireless sensor network and embedded positioning technology to achieve fast and accurate information collection and transmission. The perception layer senses the surrounding environment from all aspects. Many sensing devices actively or passively sense, collect, and transmit data to the environment of the coverage area in real-time. Many kinds of sensor nodes can be deployed in various entities, such as wearable devices, smartphones, *etc.* The sensing layer generally requires sensor devices and terminal devices to have low power consumption and high accuracy to ensure that the sensing devices remain stable in long-term operation.

The layer is located in the upper layer of the perception layer of the IoT, which is mainly responsible for the transmission, routing, and control of information. Timely and reliable transmission of received information through various Internet, public networks, private networks, mobile communication networks, telecommunication networks, *etc.* The network layer's data management and processing technology mainly includes the storage, analysis, query, further mining, and decision-making of the sensing data. The layers of the IoT can transmit all kinds of data collected by the perception layer to the application layer in real time, reliably, efficiently, and safely through various network infrastructures by the perception layer, middle layer and application layer. The sensing layer is responsible for collecting data collected by various sensor devices. These sensors can be temperature sensors, humidity sensors, pressure sensors, motion sensors, *etc.* The primary task of the perception layer is to convert information in the physical world into digital signals for further processing and transmission. Located above the perception layer, it is the key intermediate layer connecting the perception and application layers. The IoT layer uses various network infrastructures, such as local area network (LAN), wide area network (WAN), cellular network, wireless sensor network (WSN), *etc.*, to transmit various data collected by the perception layer to the application layer in real time, reliably, efficiently and securely. The application layer is located at the top of the IoT system and is the level at which users ultimately use the IoT system to obtain and process data. The application layer can be a mobile application, a Web application, a cloud service, *etc.*, to monitor, control, and analyze the data transmitted from the awareness layer. Since the interconnection with other networks and the need for comprehensive business development are not considered

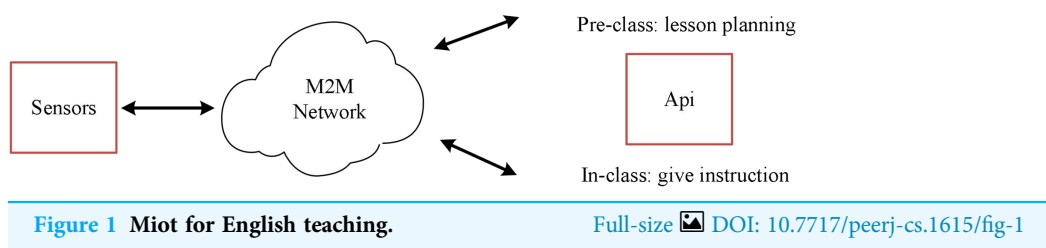

**Figure 1 Miot for English teaching.**

in the early construction of various networks, the problem of repeated network infrastructure construction is common. Therefore, the future heterogeneous Internet of Things must consider the cooperation and deep integration of multiple devices and networks.

As the interface layer between the IoT and the user, the application layer combines the IoT technology with the English course practice evaluation system. The information is coordinated across platforms and industries, and the information is analyzed, decided upon, shared, and released to maximize its availability. The application layer of the IoT mainly provides the integration and management of applications before and during English class. The application layer of the IoT further processes the perceived data. It provides services to enable users to realize the interconnection of various elements in the English course practice evaluation system..

## Evaluation of English lesson preparation content based on intelligent text analysis method

To better understand the lesson plans compiled by teachers when preparing for lessons, we evaluate the English lesson preparation content by using the intelligent text analysis method. Through the image acquisition of the English lesson preparation content through the sensor device, the image of the lesson plan content for each class is obtained. Considering that the traditional recurrent neural network has a poor effect on long-distance sentence processing and cannot capture semantic information in all directions. Although the encoding information in all orders can be obtained by bidirectional encoding, it increases the computational complexity of the model. To solve the problems and fully understand the lesson preparation content, we use the improved Transformer Encoder as the basic feature extractor, as shown in Fig. 2, which has a great advantage in calculation speed. Thanks to the self-attention mechanism, it can better capture context semantics and solve problems such as long-distance dependence. In addition, the structure of the model is improved, and the attention alignment method is used to focus on some important word vectors, weaken the influence of irrelevant words, and generate reasoning vectors with more robust semantics. In the model ending, an interaction layer is added to share semantic information between sentence pairs to form a more semantically rich word vector, which is then input to the latter layer for processing. We set up an interaction layer to achieve the fused features and extract the same information from the sentence pairs.

Each improved Transformer module contains two consecutive multi-head self-attention modules. Each module comprises Layer Normalization (LN), a multi-head self-attention

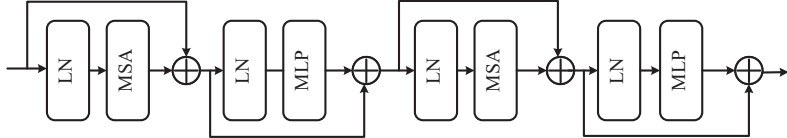

**Figure 2 Transformer block.** 

mechanism, residual connection, and Multi-Layer Perceptron (MLP). The overall calculation process is as follows:

$$\hat{z}_i = MSA(LN(z_{i-1})) + z_{i+1} \tag{1}$$
$$z_i = MLP(LN(\hat{z}_i)) + \hat{z}_i \tag{2}$$
$$\hat{z}_{i+1} = MSA(LN(z_i)) + z_i \tag{3}$$
$$z_{i+1} = MLP(LN(\hat{z}_{i+1})) + \hat{z}_{i+1} \tag{4}$$

When calculating the multi-head self-attention mechanism, each head is calculated as follows:

$$Attention(Q, K, V) = softmax\left(\frac{QK^T}{\sqrt{D}} + B\right)V \tag{5}$$

where $Q, K, V \in R^{S^2 \times D}$ refer to Query, Key, and Value matrix, $S^2$ is the number of image patches in a window, $D$ means the number of feature dimensions of the sequence; The corresponding encoding matrix is $B \in R^{S^2 \times S^2}$, whose value is obtained in the bias matrix $R^{(2S-1) \times (2S+1)}$. Self-attention allows each sequential element (such as every word in a sentence or every pixel in an image) to interact. It calculates the relevance score (or attention weight) between elements to determine how important different elements are for a particular task. This global correlation modeling enables the model to consider the relationships between all aspects simultaneously to better understand the context information in the sequence. Self-attention does not depend on the location information of the element. Each element's relevance weight to all other elements is obtained by calculating its similarity to the other elements, regardless of their actual position in the sequence. This makes the self-attention model better handle out-of-order and variation in the input sequence.

By fully extracting the visual information of the lesson plan image and establishing the semantic relationship through the encoder, we can obtain a set of semantic features. Then, the semantic features are used to design the decoder to obtain the final evaluation results of the English course lesson plan. The relevance weights of the word vectors' relative semantic features $z_{i+1}$ in GT are calculated, and the calculation process is shown in Eq. (6). The weights and each word vector are used to obtain the aligned enhanced representation vector $p_i'$.

$$p_i' = \sum_{j=1}^{n} \frac{\exp(e_{ij})}{\sum_{k=1}^{n} \exp(e_{ik})} p_i. \tag{6}$$

After calculating the model's output, the result vectors are collected and spliced together. Finally, the features for classification are extracted from the vectors to obtain the recognition results.

$$\text{logit} = \text{softmax}\left(\text{FC}\left(p_i'\right)\right) \tag{7}$$

where FC is a fully connected layer, the activation function uses the Tanh. Input $p_i'$ into a fully connected layer for linear transformation to reduce the dimension and then normalized by Softmax to obtain the predicted probability of each classification.

## Teaching intelligent scoring methods

The convolutional neural network method analyzes the context semantics and feature learning of comments, and its anti-noise and classification degrees are high. Therefore, the CNN method is selected to extract the audio and video features of teaching quality in the class. A training set was established according to the audio and video information collected by the sensor, and labels marked all samples in the training set. After automatically inputting the test dataset by convolution and pooling, the trained neural network was applied to obtain the label of the new sample, and the extraction process is shown in Fig. 3.

The K-means is a clustering algorithm. Its principle is to make the objects of different clusters as different as possible and the objects of the same cluster as much as possible according to the corresponding similarity rules. The algorithm uses distance as the criterion for dividing clusters and usually uses the Euclidean distance formula to calculate the distance between data object samples, such as the formula

$$d = \sqrt{\sum_{j=1}^{n} \left(x_i - y_i\right)^2}. \tag{8}$$

In the clustering process of the K-means clustering algorithm, the average value of all the samples in the cluster needs to be recalculated in each iteration, which is called the cluster center point. The updated calculation formula is: $c_i c_i$

$$C_i = \frac{1}{|c_i|}\sum x_i \tag{9}$$

where $C_i$ denotes a cluster.

The K-means clustering algorithm must iteratively update the divided categories and cluster centers until the termination condition is met. The default termination condition is that the model reaches the maximum or the algorithm's objective function is less than the threshold. As seen from the above description, the K-means core is the squares criterion's minimum error sum. The basic idea is to divide the given data object into the same cluster through some iterations and then recalculate the cluster center by the cyclic iterations when the criterion function converges. And obtain the output result. The k-criterion function E is defined as follows.

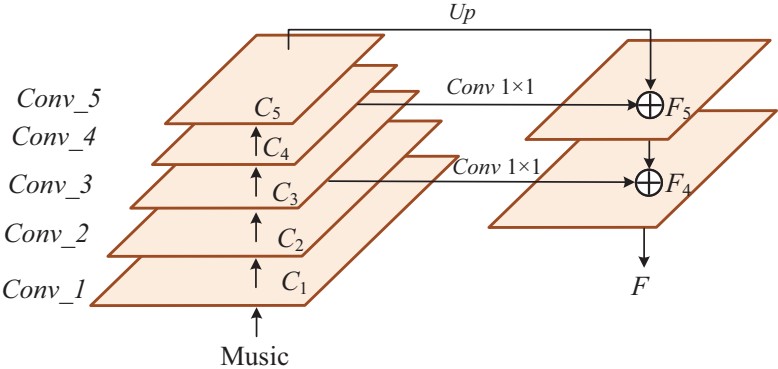

**Figure 3 The structure of extraction.**

$$E = \sum_{i=1}^{k} \sum_{x_i \in C_i} (x_i - \overline{x_i})^2.$$ (10)

The more significant E is, the lower the intra-cluster similarity is. On the contrary, the smaller E is, the higher the intra-cluster similarity is.

In the teaching intelligent scoring method, the k-means algorithm can cluster students' performance or learning behavior to classify and evaluate students. K-means is an unsupervised machine learning algorithm that divides data points into K clusters so that the data points within the same cluster are similar, and the data points between different groups are quite different. By collecting students' data in the learning process, such as answering questions, learning time, interactive behavior, *etc.*, students can be divided into several groups (clusters), each representing a kind of teaching quality. By inputting students' learning performance data into the K-means algorithm, students can be divided into different groups to assess their learning level and teaching quality according to their group. By inputting the extracted audio and video features of English teaching into the k-means algorithm model, the evaluation grade in the process of English teaching practice can be directly obtained. Among them, audio and video feature fusion refers to integrating and combining various features extracted from teaching video and audio to receive more comprehensive and accurate information to better understand the learner's behavior, emotions, and learning process. Integrating audio and video features can improve the accuracy of teaching quality recognition. Learners who watch teaching videos may have different learning behaviors, such as concentration, inattention, confusion, and so on. By integrating visual information in video and sound features in audio, students' current learning behavior can be more accurately identified, reflecting the quality of teaching from the side.

## EXPERIMENT AND ANALYSIS

### Dataset and implement details

We use the Teaching and Learning dataset to verify the effect of our evaluation system. The lesson plan is shown in Fig. 4.

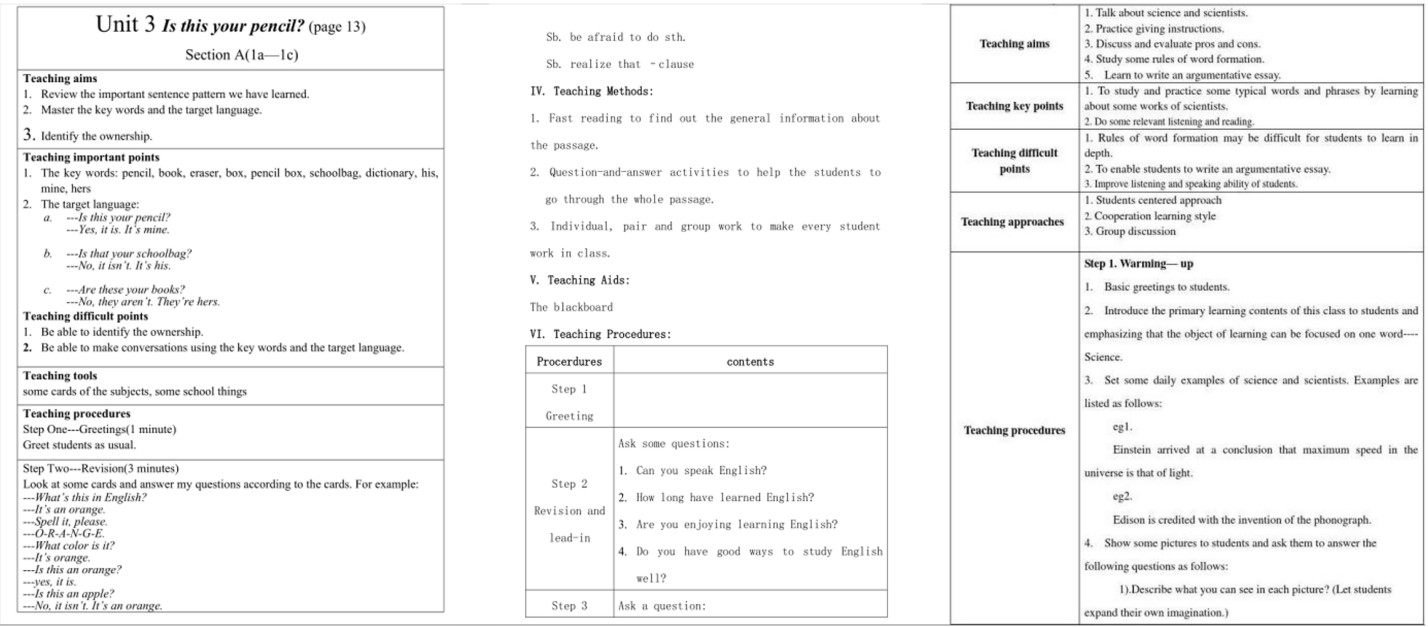

**Figure 4  The samples in the teaching and learning dataset.**

We conduct experiments on a device with an i7-12700 CPU and Rtx 3090 GPU. The operating system is Unbuntu, and the network model is implemented under the PyTorch framework. The total number of rounds of the experiment was set to 1,000, the batch size was set to 16, and the initial learning rate was set to 0.1. SGD is used as the optimizer of the model, the momentum is 0.9, and the weight decay term is set to.$1 \times 10^{-4}$. We use commonly used word segmentation tools, including NLTK, spaCy, jieba, *etc.*, to break down text in English teaching into sequences of words or sub-words that computers can understand and process. Since the length of the sentences may not be consistent, it is necessary to fill or truncate them so that all the sentences are the same length for batch processing. Finally, Word2Vec, GloVe, FastText and other methods map words to low-dimensional vector space so the model can learn word meaning and semantic information.

To evaluate the performance of our method, we use accuracy as the evaluation criterion, which is calculated as follows:

$$\text{Accuracy} = \frac{\text{TP}}{\text{TP} + \text{FP}} \tag{11}$$

where TP represents the number of correctly predicted samples, FP is the number of incorrectly predicted samples.

## Results and discussion

We conducted experiments on the evaluation methods of pre-class lesson plans for English courses on the Teaching and Learning dataset. At the same time, three evaluation grades are set up: excellent, good, and medium. Compared with three advanced technologies: Transformer (*Xiao et al., 2023*), Swin-Transformer (*Liu et al., 2022*), and Visual

Transformer (*Li et al., 2023*), the results are in Fig. 5. Our model achieves the best teaching plan content score classification performance, with an accuracy of 80.23%. Compared with the Visual Transformer, the accuracy evaluation index is improved by about 3.01%. As shown in Fig. 6, after comparing several mainstream networks with other networks that introduce attention mechanisms, such as Swin-Transformer and Visual Transformer, Our model has fewer parameters, a faster inference time, lower computational complexity, and is more lightweight. Transformer-based methods are prone to losing some features and suffer from inaccurate classification. Our model not only ensures sensitivity but also prevents the loss of features. Finally, we show the ratings made by our method for some lesson plans, as shown in Fig. 7. It can be found that our pre-class lesson plan evaluation method can accurately grade the lesson plans made by teachers. As for the content of lesson preparation, we can see that the perfect content should be shown in the leftmost sample in Fig. 7, which has a complete outline and teaching content and contains certain expansibility. In contrast, the middle teaching plan lacks these necessary contents and is vague and unclear, which is a poor teaching plan. The sample on the far right is a good-quality lesson plan that is complete in content but lacks some extension content.

In addition, the results of our methods about pre-lessons in the rating system were validated on the Coin video dataset. We select Transformer, Deit (*Touvron et al., 2021*), CLIP (*Radford et al., 2021*), and Vit to conduct comparative experiments with our model.

As shown in Table 1, the experimental results show that the CNN features obtained by FPN enhance the global learning ability of audio and video features, extract multi-level feature information, improve feature interaction ability by information fusion, aggregate all global features for deep and shallow feature deep fusion, and finally improve the accuracy of teaching quality in model recognition courses in Fig. 7. Compared to Transformer, our method improves accuracy by 4.02%. In contrast, the number of model parameters decreases by 99.22 M. Compared with Deit and Vit, our approach enhances accuracy by more than 5%, and the number of model parameters is much smaller. Compared with the more effective Vit, our method is significantly ahead in the number of model parameters, and the model accuracy is also better than CLIP. In addition, we show the training process of our classroom quality assessment model in Fig. 8. Although the k-means algorithm can help us improve the accuracy to 97.44%, there are still some drawbacks. The K-means algorithm is susceptible to selecting initial clustering centers, and different initial centers may lead to different clustering results. Each cluster is assumed to be convex; all points within the cluster are located in a circle centered around the cluster's center. This limits the effectiveness of K-means when dealing with clusters with non-convex shapes. The performance of K-means can deteriorate on high-dimensional data sets, known as the "dimensional disaster" problem. The distance between data points in high-dimensional space becomes more similar, and the clustering effect may become poor.

## Discuss and application

Through the above experiments, it can be found that our proposed English curriculum practice evaluation system, including the text intelligent analysis method of English lesson preparation content evaluation method and the teaching intelligent scoring method, can be
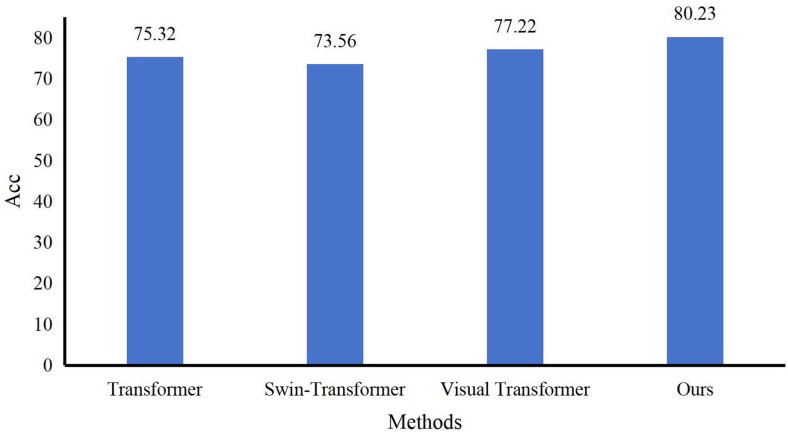

**Figure 5 Comparison with other methods.**     

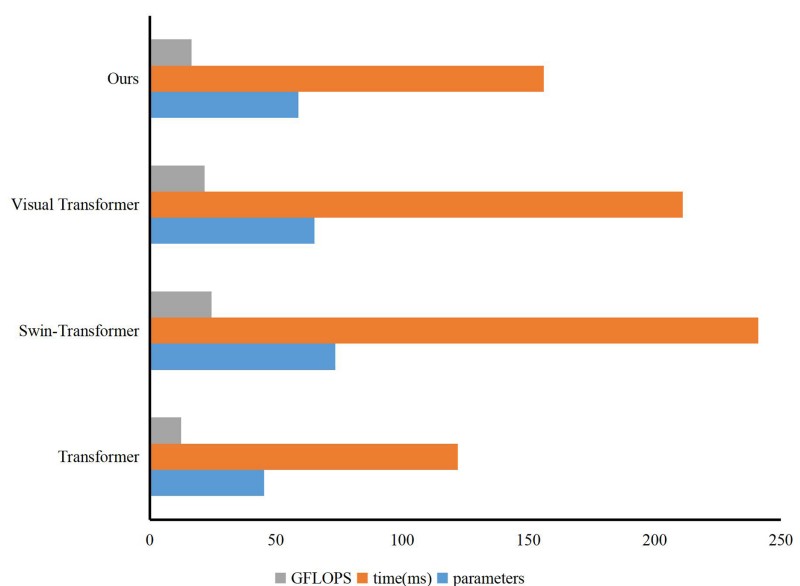

**Figure 6 The results of parameters, cost time, and GFLOPs of the methods.**
     

applied to English curriculum practice to help teachers better manage the classroom and improve the quality of teaching. Our approach utilizes natural language processing and computer vision techniques to deeply analyze English text data for a more comprehensive, objective, and accurate assessment and guidance.

The text intelligence analysis method can automatically evaluate the English teaching content of teachers' lesson preparation, which helps teachers understand the textbook's coverage degree and difficulty more quickly and saves lesson preparation time. Text intelligence analysis can be used to evaluate multiple dimensions of English lesson preparation content, such as grammar, vocabulary, semantics, difficulty, *etc.*, to find possible problems in teaching. According to the results of text intelligence analysis, personalized lesson preparation suggestions can be provided to teachers, helping them to

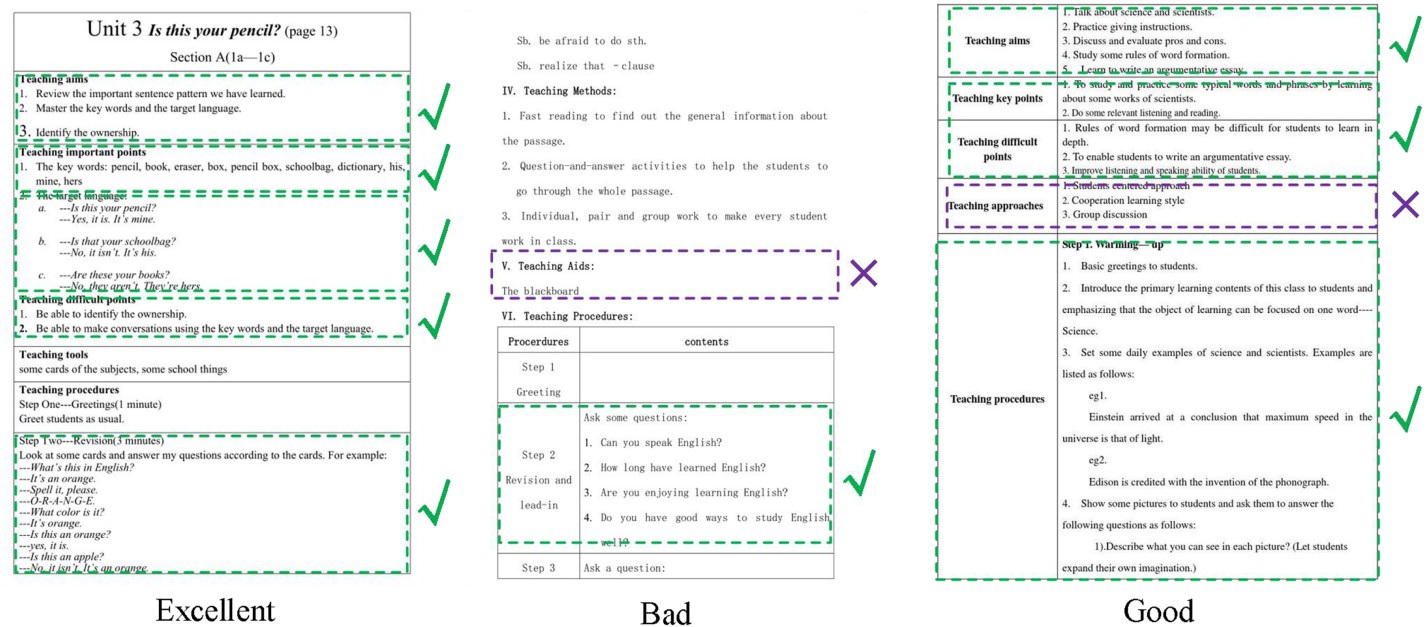

**Figure 7 The results of our methods about pre-lessons.**

**Table 1 The results of parameters and accuracy of the recognition methods.**

|  | Acc/Top1 | Parameters (M) |
| --- | --- | --- |
| Transformer | 0.9342 | 187.12 |
| Deit | 0.9239 | 190.23 |
| CLIP | 0.9728 | 234.32 |
| Vit | 0.9234 | 176.9 |
| Ours | 0.9744 | 87.9 |

better design teaching content according to the needs and levels of students. The teaching intelligent scoring method can avoid subjective bias and improve the accuracy and fairness of the evaluation results by objectively evaluating the text data of students. Teaching intelligent scoring methods can also analyze students' English writing or oral expression in real-time and provide timely feedback and suggestions to help them improve their language expression ability. With the use of instructional intelligence scoring, teachers and schools can monitor and track students' learning processes, find learning problems, and take timely teaching intervention measures. At the same time, the intelligent scoring method can provide personalized learning resources and suggestions according to the performance and level of students and help them make learning plans according to their own needs and abilities.

To give full play to the role of our English curriculum practice evaluation system, we rely on the school intranet platform and some sensors to build an English curriculum practice evaluation system integrating a multi-source mobile Internet of Things, as shown in Fig. 9.

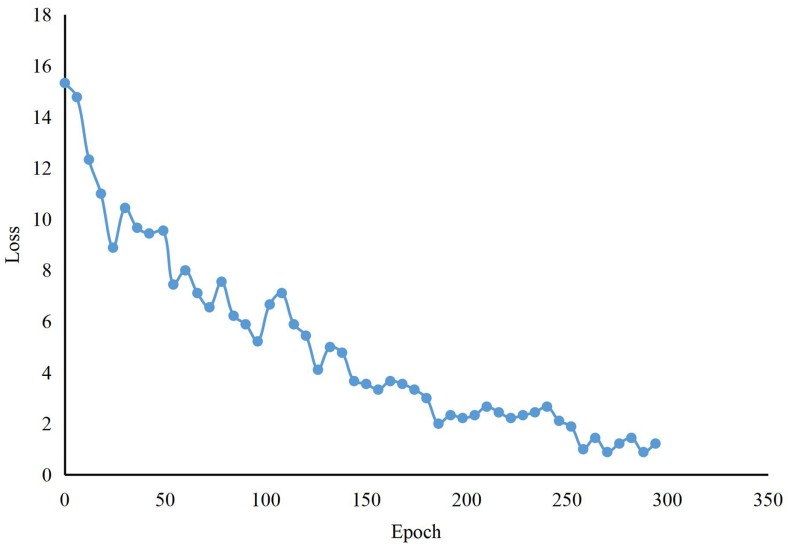

**Figure 8 The training of our methods about in-lessons.**

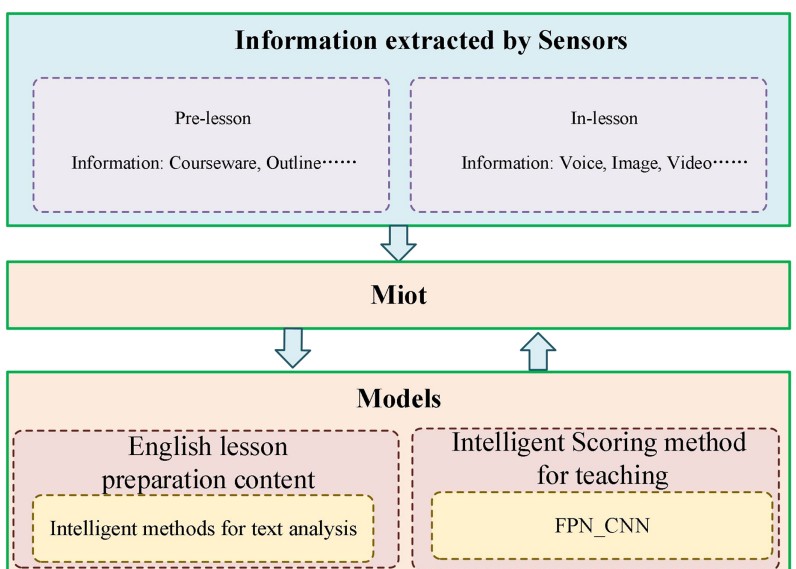

**Figure 9 English course practice evaluation system integrating multi-source Miot.**

By integrating the system into the school intranet platform, we conduct a comprehensive evaluation of English course practice according to the audio and video sensors in each classroom and the lesson preparation content of each English teacher. Firstly, the system uses sensors to collect the audio and video information of the teacher in class and sends it to the Internet of Things platform for intelligent evaluation to obtain the evaluation score in the class. Then, through the uploaded lesson preparation content, the intelligent model of the Internet of Things platform was used to rate it, and the specific score of the lesson preparation content was obtained. The score of lesson preparation

content and the evaluation score in the class were integrated to complete the evaluation of English course practice.

## CONCLUSION

To evaluate the quality of English course practice, this article discusses integrating multi-source mobile Internet of Things information technology into English curriculum practice evaluation. It proposes an English curriculum practice evaluation system based on multi-source data collection, processing, and analysis. The quality of English course preparation is evaluated by proposing a method of scoring the content of teachers' pre-class teaching plans based on text intelligence analysis. Then, CNN and FPN technology are used to rate the quality of teachers in English courses to realize the supervision of teachers in English classes. The experimental results show that our model can accurately rate the content of pre-class teaching plans and the teaching quality in class. At the same time, our model can be embedded into the school's intranet by integrating Internet of Things information technology to provide a basis for the school's subject rating. However, the k-means algorithm's limitations lead to our model's inaccuracy in handling specific samples. To solve this problem, we will focus on the study of particular situations in English curriculum practice in the future to improve the universality and stability of the model.

### Funding

The author received no funding for this work.

### Competing Interests

The author declares that they have no competing interests.

### Author Contributions

- Zhenlong Wang conceived and designed the experiments, performed the experiments, analyzed the data, performed the computation work, prepared figures and/or tables, authored or reviewed drafts of the article, and approved the final draft.

### Data Availability

The code is available in the Supplemental File.

The dataset is available at Zenodo: Sasha Cuerda. (2023). English Language Learners [Data set]. Zenodo. https://doi.org/10.5281/zenodo.7925050.

### Supplemental Information

Supplemental information for this article can be found online at http://dx.doi.org/10.7717/peerj-cs.1615#supplemental-information.

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
