# Peer review of "An English course practice evaluation system based on multi-source mobile information and IoT technology"

_PeerJ Computer Science, doi:10.7717/peerj-cs.1615_

## Round 0.1 · original submission · Major Revisions

Based on the reviewers’ comments, you may resubmit the revised manuscript for further consideration. Please consider the reviewers’ comments carefully and submit a list of responses to the comments along with the revised manuscript.

Reviewer 1 ·

Basic reporting

With the increase of online English course practice, its quality directly determines the efficiency of students in class. In order to better supervise the specific content of English courses, in order to improve the performance of English learning assessment. By analyzing the existing problems of English curriculum evaluation and the characteristics of multi-source mobile Internet of Things information technology, this paper designs a practical English curriculum evaluation system based on multi-source data collection, processing and analysis. The experiment proves that it can effectively improve the efficiency of English learning assessment, and further improve and optimize the teaching content of English education to meet the needs of actual teaching environment. In addition, this paper also has the following shortcomings:

(1) Keywords should be written in alphabetical order.
(2) The explanations about the organization of the paper are missing in the Introduction section.
(3) Current Introduction section is simple and misses some content related to the problem formulation. The Introduction does not provide relevant information for the article topic and it does not provide understandable information about the problem addressed. Furthermore, it does not provide contribution and motivation/need for such contribution.
(4) Experimental numerical results should be supplemented to highlight the effect of the model.
(5) In the related works, some Internet of Things information technology can be added to fit the theme.
(6) The author needs to introduce and explain the innovative points of self-attention mechanism to contribute.
(7) The missing Figure 1 to 9 correspond to the article description, the author needs to check and correct them.
(8) The K-means algorithm needs to be emphasized in order to highlight its practical role. What is the value of “k”?
(9) The author set three evaluation grades in Section 4.2, but they do not seem to be mentioned in the experiment.
(10) How does the information fusion used in the experiment improve the accuracy of model recognition?
(11) If possible, the author should highlight the innovative points in order to increase acceptance.
(12) Some paragraphs are too long to read. They should be divided into two or more.
(13) To increase professionalism, references should be reinforced with recent papers from reputable journals.
(14) Additional comments about the reached results should be included. Graphics and charts need more explanation.
(15) Clarifying the study’s limitations allows the readers to better understand under which conditions the results should be interpreted. A clear description of limitations of a study also shows that the researcher has a holistic understanding of his/her study. However, the authors fail to demonstrate this in their paper. The authors should clarify the pros and cons of the methods. What are the limitation(s) methodology(ies) adopted in this work? Please indicate practical advantages, and discuss research limitations.
(16) Some more recommendations and conclusions should be discussed about the paper considering the experimental results. The Conclusion section is weak. Furthermore, there is not any discussion section about the results. It should briefly describe the findings of the study and some more directions for further research. The authors should describe academic implications, major findings, shortcomings, and directions for future research in the conclusion section.

Experimental design

(1) Keywords should be written in alphabetical order.
(2) The explanations about the organization of the paper are missing in the Introduction section.
(3) Current Introduction section is simple and misses some content related to the problem formulation. The Introduction does not provide relevant information for the article topic and it does not provide understandable information about the problem addressed. Furthermore, it does not provide contribution and motivation/need for such contribution.
(4) Experimental numerical results should be supplemented to highlight the effect of the model.
(5) In the related works, some Internet of Things information technology can be added to fit the theme.
(6) The author needs to introduce and explain the innovative points of self-attention mechanism to contribute.
(7) The missing Figure 1 to 9 correspond to the article description, the author needs to check and correct them.
(8) The K-means algorithm needs to be emphasized in order to highlight its practical role. What is the value of “k”?
(9) The author set three evaluation grades in Section 4.2, but they do not seem to be mentioned in the experiment.
(10) How does the information fusion used in the experiment improve the accuracy of model recognition?
(11) If possible, the author should highlight the innovative points in order to increase acceptance.
(12) Some paragraphs are too long to read. They should be divided into two or more.
(13) To increase professionalism, references should be reinforced with recent papers from reputable journals.
(14) Additional comments about the reached results should be included. Graphics and charts need more explanation.
(15) Clarifying the study’s limitations allows the readers to better understand under which conditions the results should be interpreted. A clear description of limitations of a study also shows that the researcher has a holistic understanding of his/her study. However, the authors fail to demonstrate this in their paper. The authors should clarify the pros and cons of the methods. What are the limitation(s) methodology(ies) adopted in this work? Please indicate practical advantages, and discuss research limitations.
(16) Some more recommendations and conclusions should be discussed about the paper considering the experimental results. The Conclusion section is weak. Furthermore, there is not any discussion section about the results. It should briefly describe the findings of the study and some more directions for further research. The authors should describe academic implications, major findings, shortcomings, and directions for future research in the conclusion section.

Validity of the findings

(1) Keywords should be written in alphabetical order.
(2) The explanations about the organization of the paper are missing in the Introduction section.
(3) Current Introduction section is simple and misses some content related to the problem formulation. The Introduction does not provide relevant information for the article topic and it does not provide understandable information about the problem addressed. Furthermore, it does not provide contribution and motivation/need for such contribution.
(4) Experimental numerical results should be supplemented to highlight the effect of the model.
(5) In the related works, some Internet of Things information technology can be added to fit the theme.
(6) The author needs to introduce and explain the innovative points of self-attention mechanism to contribute.
(7) The missing Figure 1 to 9 correspond to the article description, the author needs to check and correct them.
(8) The K-means algorithm needs to be emphasized in order to highlight its practical role. What is the value of “k”?
(9) The author set three evaluation grades in Section 4.2, but they do not seem to be mentioned in the experiment.
(10) How does the information fusion used in the experiment improve the accuracy of model recognition?
(11) If possible, the author should highlight the innovative points in order to increase acceptance.
(12) Some paragraphs are too long to read. They should be divided into two or more.
(13) To increase professionalism, references should be reinforced with recent papers from reputable journals.
(14) Additional comments about the reached results should be included. Graphics and charts need more explanation.
(15) Clarifying the study’s limitations allows the readers to better understand under which conditions the results should be interpreted. A clear description of limitations of a study also shows that the researcher has a holistic understanding of his/her study. However, the authors fail to demonstrate this in their paper. The authors should clarify the pros and cons of the methods. What are the limitation(s) methodology(ies) adopted in this work? Please indicate practical advantages, and discuss research limitations.
(16) Some more recommendations and conclusions should be discussed about the paper considering the experimental results. The Conclusion section is weak. Furthermore, there is not any discussion section about the results. It should briefly describe the findings of the study and some more directions for further research. The authors should describe academic implications, major findings, shortcomings, and directions for future research in the conclusion section.

Additional comments

There is not any general comments not covered by the three areas above. Thank you...

·

Basic reporting

In recent years, industries have continued to adopt Internet of Things (IoT) technology, with very considerable scenario adaptability. In order to better monitor the specific content of English courses, this paper discusses how to apply multi-source mobile Internet of Things information technology to the practical evaluation system of English courses to improve the performance of English learning assessment. This paper designs a practical English course evaluation system based on multi-source data collection, processing and analysis, and finally proves that it can effectively improve the efficiency of English learning evaluation. The following changes are required for this article to be accepted by the journal.
1. Abstract keywords should be rewritten according to the abstract to highlight the model approach;
2. In the last paragraph of the introduction, the author should arrange the contributions by points;
3. Some of the expressions are not accurate. For example, in line 137 "Then the collected information is transmitted accurately in real time on the basis of adapting to various heterogeneous networks and protocols ", is the information collected here in real-time?
4. The author should add a paragraph to each section to explain the main content;

Experimental design

5. As for the improved Transformer Encoder mentioned in section 3.1, what specific improvements do we need to know;
6. What is the effect of adding an interaction layer at the end of the model?
7. Data set preprocessing and parameter settings should be described in detail;

Validity of the findings

8. In order to enhance the validity of this model, ablation experiments should be added to the experimental part;
9. There is too little content in the conclusion, the author should add more content for the reader to understand.

---

## Round 0.2 · Minor Revisions

The reviewers are satisfied with the revised version and the technical contributions of the manuscript.

However, the paper needs considerable improvement in the presentation and writing. It is advised to thoroughly re-review the paper text and correct all the language-related issues.

**Language Note:** The Academic Editor has identified that the English language must be improved. PeerJ can provide language editing services - please contact us at [email protected] for pricing (be sure to provide your manuscript number and title). Alternatively, you should make your own arrangements to improve the language quality and provide details in your response letter. – PeerJ Staff

Reviewer 1 ·

Basic reporting

I already review this paper and the current version has improved with respect to the previous one.

Experimental design

The authors did the required changes and enhanced the work as required.

Validity of the findings

The reviewer has carefully evaluated the authors' responses and their actions in the revised version of the paper. Found them persuasive and applicable. Hence, the revised paper is accepted and merits publishing.

Additional comments

The reviewer has carefully evaluated the authors' responses and their actions in the revised version of the paper. Found them persuasive and applicable. Hence, the revised paper is accepted and merits publishing.

·

Basic reporting

no comment

Experimental design

no comment

Validity of the findings

no comment

---

## Round 0.3 · accepted · Accept

The revisions are satisfactory and your paper is recommended for publication, congratulations.